# The Effects of Commonly Consumed Dietary Fibres on the Gut Microbiome and Its Fibre Fermentative Capacity in Adults with Inflammatory Bowel Disease in Remission

**DOI:** 10.3390/nu14051053

**Published:** 2022-03-02

**Authors:** Konstantinos Gerasimidis, Ben Nichols, Mhairi McGowan, Vaios Svolos, Rodanthi Papadopoulou, Margarita Kokkorou, Martina Rebull, Teresita Bello Gonzalez, Richard Hansen, Richard Kay Russell, Daniel Richard Gaya

**Affiliations:** 1Human Nutrition, School of Medicine, College of Medical, Veterinary and Life Sciences, University of Glasgow, New Lister Building, Glasgow Royal Infirmary, Glasgow G31 2ER, UK; ben.nichols@glasgow.ac.uk (B.N.); mhairi_mcgowan@msn.com (M.M.); vaios.svolos@glasgow.ac.uk (V.S.); rodanthi.papadopoulou@gmail.com (R.P.); m.kokkorou@gmail.com (M.K.); martinarebull@gmail.com (M.R.); teresita.bellogonzalez@wur.nl (T.B.G.); 2Department of Paediatric Gastroenterology, Hepatology and Nutrition, Royal Hospital for Children, Glasgow G51 4TF, UK; richard.hansen@ggc.scot.nhs.uk; 3Department of Paediatric Gastroenterology, Hepatology and Nutrition, Royal Hospital for Children, Edinburgh EH9 1LF, UK; richard.russell@nhslothian.scot.nhs.uk; 4Gastroenterology Unit, Glasgow Royal Infirmary, Glasgow G31 2ER, UK; daniel.gaya@ggc.scot.nhs.uk

**Keywords:** inflammatory bowel disease, microbiome, fibre, short-chain fatty acids, fermentation

## Abstract

Introduction: It has been suggested that the gut microbiome of patients with inflammatory bowel disease (IBD) is unable to ferment dietary fibre. This project explored the in vitro effect of fibre fermentation on production of short-chain fatty acids (SCFA) and on microbiome composition. Methods: Faecal samples were collected from 40 adults (>16 y) with IBD (*n* = 20 with Crohn’s disease and *n* = 20 with ulcerative colitis) in clinical remission and 20 healthy controls (HC). In vitro batch culture fermentations were carried out using as substrates maize starch, apple pectin, raftilose, wheat bran, α cellulose and a mixture of these five fibres. SCFA concentration (umol/g) was quantified with gas chromatography and microbiome was profiled with 16S rRNA sequencing. Results: Fibre fermentation did not correct the baseline microbial dysbiosis or lower diversity seen in either patients with CD or UC. For all fibres, up to 51% of baseline ASVs or genera changed in abundance in HC. In patients with IBD, fermentation of fibre substrates had no effect on species or genera abundance. Production of SCFA varied among the different fibre substrates but this was not different between the two IBD groups and compared to HC after either 5 or 24 h fermentation. Conclusions: Despite extensive microbial dysbiosis, patients with IBD have a similar capacity to ferment fibre and release SCFA as HC. Fibre supplementation alone may be unlikely to restore to a healthy status the compositional shifts characteristic of the IBD microbiome.

## 1. Introduction

Several studies have explored the role of the gut microbiome or host diet on the course of inflammatory bowel disease (IBD) [1], but few have explored their interaction [2,3]. A dietary component which has attracted major scientific and public interest in the aetiology and management of IBD is fibre [4]. In healthy people, fermentation of dietary fibre regulates microbial growth and composition, and the end-products of this anaerobic saccharolytic process are predominantly short-chain fatty acids (SCFA) [5]. SCFA are known to exert major health-promoting effects. Acetate is an energy substrate to the host via de novo lipogenesis; propionate is implicated in glucose homeostasis and appetite regulation; and butyrate is the preferable energy substrate for colonocytes, regulating local inflammatory responses and inducing apoptosis in transformed cells [5]. Dietary substrate availability is the major determinant of both the amount and type of SCFA produced, with both microbiome composition and the luminal environment being important co-factors [6]. Despite the importance of SCFA in preserving health, their role in IBD remains unclear and is less well studied [4], in direct contrast with our extensive and increasing knowledge of the composition of the microbiome in IBD and its role in disease development.

Previous reports of the faecal concentration of SCFA, as proxies of luminal production, in patients with IBD have produced inconsistent findings and, paradoxically in children with active Crohn’s disease (CD), levels decrease with disease improvement following induction therapy with exclusive enteral nutrition [2,7]. Explaining these inconsistencies is challenging but factors may include: inter-individual variation in measurements (including the impact of faecal volume and frequency on quantitating SCFA), differing dietary habits in patients with IBD and gut transit time [8]. As the majority of SCFA are produced in the caecum and utilised locally, residual faecal concentrations of SCFA can only be considered an approximate measurement of true production, even when meticulous 72 h stool collections are performed. In an elegant study, James et al. showed that despite a tendency to lower habitual fibre intake in ulcerative colitis (UC) patients, faecal non-starch polysaccharide and starch concentrations were unexpectedly seen to be threefold higher than in healthy controls (HC) [9]. Perhaps most interestingly, concentrations of SCFA were similar, potentially suggesting altered utilisation. The authors explored this hypothesis by increasing resistant starch and wheat bran intake in patients with UC but this did not increase faecal fermentation patterns or influence the composition of the microbiome. They concluded that luminal fermentation is diminished in patients with UC due to abnormal functioning of the gut microbiome [9]. These observations, along with lack of efficacy seen in prebiotic and fibre supplementation studies to positively influence clinical outcomes in patients with IBD [10], may suggest a functional impairment of the gut microbiome in its response to dietary fibre and, consequently its health-promoting effects to the host. Encouragingly, pharmacological administration of SCFA to the gut has been shown to improve markers of disease activity in patients with IBD [11,12] suggesting that the lack of efficacy attributed to dietary fibre may be due to its inability to be fermented and release beneficial SCFA.

While in vitro fermentation studies, using faecal inoculum, are not an exact simulation of human gut physiology, they are particularly useful in comparative studies when the net production of bacterial metabolites and changes in bacterial composition are the outcome of interest [13]. They offer the flexibility to standardise many experimental conditions, such as the amount of microbial community used, luminal redox and pH, and gut transit time which are unable to be performed in vivo. They also allow the study of different fermentation conditions or fermentable substrates using the same faecal inoculum, minimising inter-sample variation even from samples collected from the same individual.

Altogether, evidence suggests that patients with UC are unable to ferment fibre to the same extent as healthy people. This probably explains why fibre intervention studies fail to improve clinical outcomes, despite the potential clinical effectiveness of therapeutic doses of SCFA. Here, we set up an in vitro fermentation study to explore the comparative effect of five dietary fibres, commonly found in the Western diet, on the composition of the gut microbiome and production of SCFA in patients with IBD and HC.

## 2. Materials and Methods

### 2.1. Participants

This was an adult study which recruited to three groups: Crohn’s disease and UC in clinical remission, and HC. Recruitment of patients with IBD took place at Glasgow Royal Infirmary clinics and recruitment of HC was carried out via advertisement in the broader area of Glasgow. Clinical remission was ascertained by the attending consultant as a Harvey Bradshaw index [14] ≤ 5 and partial Mayo score [15] ≤ 1 during their clinic appointment. None of the participants used antibiotics three months prior to sample collection. All participants provided written informed consent.

We undertook two distinct in vitro fermentation experiments. Firstly, we fermented different types of dietary fibre using the faecal material of our participants and described changes in gut microbiome composition after fermentation. Secondly, we utilised a similar experimental design to describe the production of SCFA after fermentation.

### 2.2. In Vitro Batch Fermentations

The whole bowel movement was collected from the participant’s residence using a collection kit, stored immediately under cold anaerobic conditions (Oxoid AnaeroGen Sachet; Thermos Scientific, Manchester, UK) and transferred to the laboratory to be processed within 4 h of defecation, with a mean (SD) collection and processing time of 2.1 ± 0.1 h.

The batch fermentations methodology used was described previously [13]. From each donor, a faecal slurry (16% *w/v*) was prepared using 16 g of faecal matter homogenised in 100 mL Sorensen’s buffer pH 7, boiled and degassed under oxygen-free nitrogen stream. The faecal slurry was strained through 30-denier nylon stockings to remove coarse material and maintained in suspension by continuous agitation using a magnetic stirrer. In a 150 mL flask, 5 mL of 16% faecal slurry was added along with 42 mL of in-house prepared fermentation medium, 2 mL of reducing solution and 500 mg of fibre substrate as described below. This assumes that an average person has a faecal output of 120 g/day [16] and a recommended intake of the fibre of 30 g/day, would roughly double the amount of fibre available for fermentation per gram of faeces.

The fermentation medium was prepared in-house (1 L). It consisted of 225 mL of macromineral solution (0.04 M Na_2_HPO_4_, 0.046 M KH_2_PO_4_, and 0.002 M MgSO_4_·7H_2_O), 225 mL buffer solution (0.051 M NH_4_HCO_3_ and 0.417 M NaHCO_3_), 112.5 μL of micromineral solution (0.898 M CaCl_2_·2H_2_O, 0.505 M MnCl_2_·4H2O, 0.042 M CoCl_2_·6H2O, and 0.296 M FeCl_3_·6H_2_O), 1.125 mL of 0.1% resazurin solution, 450 mL of 5 mg/mL Tryptone and 76 mg of mixed bile extract from porcine. Once the solution was made, it was boiled, 100 mg of sterile mucin from porcine stomach was added, degassed under oxygen-free nitrogen, and adjusted to pH 7 to mimic the distal intestinal environment. Reducing solution (50 mL) was made up of 2 mL of 1 M NaOH, 312.5 mg of cysteine hydrochloride and 312.5 mg of Na_2_S·9H_2_O.

The fibre substrates used were 500 mg of apple pectin (Sigma-Aldrich, 76282, Gillingham, UK); raftilose (Orafti^®^ P95, Beneo™, Mannheim, Germany); α-cellulose (Sigma-Aldrich, C8002, Gillingham, UK); high resistant maize starch (HI-MAIZE^®^, National Starch and Chemical Ltd., Manchester, UK); and wheat bran. We chose these fibres as representative of food consumed in the Western diet [17]. Additionally, 100 mg of each of the fibres above were pooled to provide 500 mg of mixed fibres to be included in the batch culture fermentations. A non-substrate control (NSC), without the addition of a fibre substrate, was used for each participant.

Seven fermentation flasks, one for each fibre, the mix of all fibres and the NSC were degassed under oxygen-free nitrogen stream and incubated in a shaking water bath at 37 °C at 60 strokes/min. A baseline sample was collected from the NSC prior to incubation. For the study of the effects of fibres on microbiome composition we collected aliquots at 0 and after 48 h fermentation. Fermentation slurry aliquots were stored at −80 °C and total DNA was extracted within a month of collection.

For the second experiment, where we evaluated the fermentative capacity of the gut microbiome, fermentation aliquots were drawn at 5 and 24 h of fermentation to assess the rate (e.g., speed) of production of SCFA and total amount produced, respectively. Aliquots of fermentation slurry for SCFA analysis were collected and stored in a 3:1 ratio with 1 M NaOH at −20 °C until analysis.

### 2.3. Genomic DNA Extraction

Extraction of genomic DNA was performed for the NSC at 0 h (starting inoculum) and for all fibre substrates at 48 h using the chaotropic method [18], in freeze-dried fermentation slurries. Genomic DNA quality and quantity were evaluated on 1% agarose gel, with NanoDrop^TM^ and Qubit^TM^ assays.

### 2.4. Quantification of Bacterial Load and 16S rRNA Sequencing

Real-time quantitative PCR was carried out for quantification of the total bacterial load (16S ribosomal RNA gene copies per ml of slurry) of the starting inoculum using TaqMan chemistry on a 7500 Real-Time PCR System (Applied Biosystems, Carlsbad, CA, USA) as described previously [2]. Serial dilution of *Blautia coccoides* was used as standards for absolute quantification. All reactions were run in triplicate. For microbiome profiling the V4 region of the 16S rRNA gene was sequenced on a MiSeq using 2 × 250 bp paired-end reads [19]. Microbiome profiling was performed in all fibres with the exception of α cellulose.

### 2.5. SCFA Analysis

At both 5 and 24 h fermentation points, 4.5 mL aliquots were collected from the flasks and added to 1.5 mL 1 M NaOH and stored at −20 °C until SCFA analysis. For SCFA and branched-chain fatty acid (BCFA) extraction, 800 µL of the fermentation slurry was extracted. Orthophosphoric acid (100 µL) and 100 µL internal standard (2-ethyl butyric acid in 2 M NaOH; 73.8 mM) were added and mixed by vortexing. To this mixture, 1 mL of diethyl-ether was added and vortexed for 1 min. The ether phase was removed, and the ether extraction was repeated twice more with the three extracts pooled, stored in gas-tight vials and analysed by gas chromatography with flame ionisation detection (7820A GC System, Agilent Technologies LDA, Cheadle, UK). SCFA concentrations were calculated using the ratio of each SCFA to the internal standard (73.8 mM, 2-ethylbutyric acid). The gas chromatographer was calibrated for individual SCFA responses against external authentic standards [acetate (185.825 mM), propionate (144.447 mM), butyrate (114.189 mM), valerate (83.4348 mM), hexanoate (76.522 mM), heptanoate (65.7951 mM), octanoate (53.178 mM), isobutyrate (97.3056 mM), isovalerate (87.0342 mM) and isohexanoate (52.6385 mM), all in 2 M NaOH].

### 2.6. Faecal Calprotectin and Water Content

Faecal calprotectin (FC) concentration was measured using the ELISA kit (EDI TM Quantitative Fecal Calprotectin ELISA, KT-849, Epitope Diagnostics, Inc, San Diego, CA, USA.) according to the manufacturer specifications. Raised faecal calprotectin was defined as a measurement >250 μg/g. Faecal water content (%) was calculated by weighing the samples before and after lyophilisation under vacuum.

### 2.7. Bioinformatics

Amplicon sequence variants (ASVs) were generated from the raw data using the dada2 pipeline [20]. Quality filtering was performed with a maximum expected error value for merged sequences of 2, and sequences were truncated at the first instance of a quality score less than 2. The core dada2 algorithm was applied to remove sequencing noise, forward and reverse sequences were merged, and an ASV table was generated. Chimeras were removed using the dada2 de novo method and sequences longer than 260 bp and shorter than 245 bp were filtered out. Repeated samples were aggregated and samples with fewer than 5000 reads were excluded from analysis. ASVs were taxonomically classified to the genus level against the SILVA 16S reference dataset, release 132 [21], using the assignTaxonomy function in dada2. Data from this study may be made available to third party on request.

### 2.8. Statistical Analysis

All statistical analyses were carried out in R version 4.0.2. The Benjamini–Hochberg correction method was applied whenever multiple statistical tests were used, and adjusted *p*-values are reported in the results. Multiple Kruskal–Wallis tests were used to assess the effect of different conditions and fibres on the change in SCFA. Analysis of microbial community structure and diversity was carried out using the vegan [22] and phyloseq [23] packages in R in addition to some in-house developed code. Alpha-diversity was assessed using the Chao1 richness estimate, Shannon diversity index, and Pielou’s evenness. The Adonis function in the vegan package was used for all applications of permutation ANOVA to assess the influence of categorical variables on community structure. Pairwise analysis was carried out by repeating this method for all pairwise combinations, adapted so that permutations could be constrained by strata. This allowed paired comparisons to be drawn between factors taking into account participant ID.

To determine differentially abundant ASVs, the least abundant ASVs, together comprising less than 10% of all reads, were first removed. Significantly changing ASVs/genera were identified using paired Wilcoxon tests on the log-proportional abundances of each ASV/genus at 0 and 48 h.

### 2.9. Ethical Considerations

This study gained ethical approval from the NHS West of Scotland Research Ethics Committee 4 with respect to patient participation (Ref: 17/WS/0207), and the University of Glasgow, College of Medical, Veterinary and Life Sciences (MVLS) Ethics Committee with respect to HC (Ref: 200130161).

## 3. Results

### 3.1. Effect of Fibre on Microbiome Composition

Thirty-six adults of Caucasian origin provided samples. Most patients with CD had disease involving the ileum and those with UC had extensive colonic disease (Table 1).

There was no difference in faecal water content (%), a proxy of diarrhoea, between the three participant groups (Appendix A).

From these 36 participants, 249 unique samples were sequenced for microbiome analysis at baseline and following fermentation. Fibre fermentation influenced the baseline microbiome structure of all fibres and conditions, except for samples of patients with CD undergoing fermentation with raftilose (Figure 1). This effect was more pronounced for HC than patients with UC or CD, and less in patients with UC compared to patients with CD, and for each fibre separately (Figure 1). The NSC and wheat bran influenced microbiome structure the most, particularly in HC, and the mixed fibre substrate the least (Figure 1).

Fibre fermentation did not correct the baseline dysbiosis seen in people with CD or UC (Figure 2). Microbial communities (β diversity) clustered according to both participant group and fermentable substrate. In aggregated analysis, where the average microbiome structure for each fibre and condition was displayed, 43% (*p* < 0.0001) and 42% (*p* < 0.0001) of the variability in mean community structure was explained by condition and substrate, respectively (Figure 2).

Prior to fermentation, the microbiome of patients with CD and UC presented a significantly lower Chao1 richness and Shannon α diversity compared to HC. Largely, these microbiome features remained uninfluenced following fermentation of the fibre substrates and the NSC (Figure 3). In HC, fermentation decreased Chao1 richness and the Shannon α diversity index for the fermentable fibres HI-MAIZE^®^, apple pectin, raftilose and mixed fibre, but not for the NSC and the less fermentable wheat bran (Appendix A). Similar effects were observed for Shannon α diversity for samples from UC patients and patients with CD, although the Chao1 richness estimate was not influenced by fibre fermentation in samples from patients with CD. There was no evidence of differences in diversity or richness indices between samples of patients with CD and UC for all fibres studied (Figure 3).

### 3.2. Effect of Fibre Fermentation on Taxon Relative Abundance

Prior to fermentation, significant differences in the relative abundance of ASVs were observed between patients with UC or CD and HC. In comparison to HC, patients with CD and UC had a significantly lower abundance of ASVs from Firmicutes, the majority of which are known fibre fermenters (Appendix A). For all fibres, a higher proportion of ASVs or genera changed in abundance in HC compared to the other two groups, in which fibre substrates had no effect (Figure 4). In HC, between 11 and 51% of all ASVs changed following fermentation with all fibres other than apple pectin. Interestingly, the abundance of fewer ASVs changed with the mixed fibre substrate than with any of the other fibres (Figure 4 and Appendix A). In HC, the effect of each fibre on the abundance of taxa was specific to each type of fibre. Raftilose and HI-MAIZE^®^ stimulated the growth of ASVs belonging to Firmicutes, most of which were fibre fermenters. The same two fibres and wheat bran reduced the abundance of *Escherichia coli.* For wheat bran, a diminishing effect was also observed for ASVs of *Alistipes* and *Bacteroides* too (Figure 4). Linear discriminant analysis effect size produced similar results as with the original analysis (Appendix A).

Except for the NSC in samples from patients with UC, fermentation of the various fibres for 48 h did not influence the abundance of any ASVs or genera in the groups of patients with IBD. For example, in contrast to the >35% baseline change to ASVs in HC in response to HI-MAIZE^®^ and raftilose, strikingly no change was observed for any ASV in patients with UC or CD (Figure 4). These findings remained even when analysis was performed after filtering out ASVs which were not present in all participant group-fibre combinations.

### 3.3. Effect of Fibre Fermentation on Fibre Fermentation and Production of SCFA

A total of 24 other participants provided samples to test the in vitro fibre fermentative capacity of the gut microbiome of patients with IBD in clinical remission against HC (Table 1). Most of the patients had CRP (<7 mg/L) and FC (<250 μg/g) levels within the normal range. Ileocolonic disease and extensive colonic disease were the most common disease location in patients with CD and UC at disease diagnosis, respectively (Table 1). There was no difference between the three participant groups in the total bacterial load per ml of faecal slurry used in in vitro fermentation experiments (Median, IQR, HC: 7.5 [7.4:7.7] vs. CD: 7.6 [7.4:7.8] vs. UC: 7.6 [7.4:7.7] copies of 16S rRNA gene copies/mL of faecal slurry; *p* = 0.929), suggesting that production of SCFA was unbiased by variation of the bacterial load in the starting inoculum. Prior to fermentation, there were no statistically significant differences in the levels of SCFA in faeces between the three groups (Appendix A).

Production of SCFA was quantified in a total of 336 fermentation samples. The amount and type of SCFA produced varied significantly among participants but also according to the type of fibre substrate fermented. In pooled analysis (irrespective of participant group), fermentation of apple pectin produced the highest concentration of acetate (Figure 5). Propionate production was the highest for apple pectin, HI-MAIZE^®^, mixed fibre and wheat bran without any significant differences between them. Butyrate production which was the highest for HI-MAIZE^®^ and raftilose followed by apple pectin and mixed fibre. Wheat bran produced the highest amount of valerate and hexanoate amongst all fibres whereas increased production of the BCFA, iso-valerate and iso-butyrate, were observed for the least fermentable fibres α cellulose, wheat bran and the NSC (Figure 5).

Production of SCFA was not significantly different between the three participant groups, with a few exceptions. Following 5 h of fermentation, samples from patients with UC produced more butyrate from HI-MAIZE^®^, and samples from patients with CD more hexanoate from pectin and HI-MAIZE^®^ than HC (Appendix A). However, these signals were lost following adjustment for multiple testing.

At 24 h of fermentation, the higher production of hexanoate from pectin by the samples of patients with CD remained statistically significant whereas fermentation of HI-MAIZE^®^ from samples of patients with UC produced significantly lower levels of heptanoate and the BCFA isobutyrate and isovalerate (Figure 6). The effect of HI-MAIZE^®^ on production of SCFA remained significant after correction for multiple testing (Figure 6).

## 4. Discussion

The role of fibre and SCFA remains unclear in the aetiology of IBD and its management. Previous randomised controlled trials using either prebiotics or fibre supplementation have had inconsistent success in influencing disease activity in UC, and no benefit in CD [1,10]. Likewise, treatment with exclusive enteral nutrition in children with active CD diminishes the concentration of faecal SCFA with, however, a parallel improvement in disease activity and amelioration of FC levels [2]. Albeit counterintuitive or disappointing, such results suggest that either the role of these dietary components is not important in the course of IBD or these microbial therapeutics fail to achieve their presumed primary objective to favourably modify the gut microbiome and subsequently produce a clinical benefit. Here, we show that in patients with IBD in clinical remission, five different types of rapid, slow and minimally fermentable fibre failed to correct IBD-related microbiome dysbiosis or to influence the composition of bacterial taxa. This was in direct contrast to HC, where extensive changes in species abundance were observed after fibre fermentation. It is therefore possible that fibre supplementation might not be a successful strategy to promote colonisation of beneficial commensals over pathobionts and other strategies might be required to reverse microbiome dysbiosis in IBD and improve disease outcomes [24]. Such interventions may include eradication of a dysbiotic microbiome with antibiotics and recolonisation with faecal material transfer from healthy donors, which has shown promise in the treatment of patients with active UC [25]. However, it is also possible that lack of an effect in this study is due to a higher inter-individual variation in the microbial composition of patients with IBD than HC, at least at the ASV or the genus level. This suggests that the same intervention may have variable effects to patients or no effects at all, and points towards stratified nutritional therapy. More work is also needed to explore a broader range of IBD phenotypes and levels of disease activity before we consider whether or not stratified nutritional therapy in IBD.

Nonetheless, the capacity of the gut microbiome of patients with IBD to ferment fibre substrates and produce SCFA was maintained and no major differences were observed between the three participant groups, and for each fibre separately. Compared to HC, patients with CD and UC in remission showed similar capacity to ferment dietary fibres and produced similar concentrations of SCFA. Likewise, the speed of fermentation, as this was assessed following 5 h of fermentation, did not differ between the three groups. Thus, the initial hypothesis that the gut microbiome of patients with IBD in remission is unable to ferment fibre was rejected, at least in a controlled in vitro experiment and in isolation from any host effect. Collectively this finding suggests that although fibre fermentation is unable to alter the microbiome composition of patients with IBD, their collective functional capacity is preserved. This may indicate an ability of the gut microbiome of patients with IBD in remission to adapt, become more efficient and use multiple metabolic pathways in promoting SCFA production. In turn, this was previously demonstrated by the fact that increased intake of oat bran increase faecal butyrate production [26].

The current study also provides data regarding which types of fibre produce a specific profile and amount of SCFA, hence allowing the development of future targeted dietary intervention. This study is, however, unable to answer whether absorption and metabolism of generated SCFA is similar in patients with IBD compared with HC. However, the absence of differences in net in vitro production of SCFA between the three groups coupled with the fact that the residual levels of SCFA in faeces did not differ either offer confidence that neither production nor absorption of SCFA is impaired in adults with CD and UC in clinical remission. Previous research also found no difference in the rate of metabolism of butyrate between patients with quiescent UC and HC, collectively suggesting that patients with IBD are unlikely to have a primary metabolic defect of butyrate metabolism [27].

Clinical trials have previously measured the concentration of SCFA in faeces of patients with IBD but results in the literature remain broadly inconsistent. Takaishi and colleagues reported a significant reduction in the faecal concentrations of propionate and butyrate in 39 UC patients and 12 with CD compared to 10 HC [28]. In contrast, Nemoto showed that patients with UC in remission had a lower concentration of SCFA than HC [29]. An Indian study described significant decreases in the concentration of butyrate and acetate in the faecal samples of active UC patients compared to the control group, but no significant differences were found between the controls and those with quiescent UC, suggesting disease activity might be an important modifier [7]. We have also previously shown no difference in the concentration of SCFA between children with active CD or CD in remission and when compared to HC [2]. It is possible that inconsistent results reflect the challenges in unveiling primary disease processes from secondary effects on host physiology (e.g., gut motility) and the luminal ecosystem; variability in gut inflammation; and the significant confounders of pharmacological and dietary differences between individual participants and different studies in centres including those in different countries. All of this may directly or indirectly influence the gut microbiome, the composition of which remains a fundamental component of fermentation.

There are some limitations in the current study. The sample size was modest and different groups of participants were enrolled in the two sub-studies. Although batch faecal fermentations are very useful tools for short-term studies and allow the study of the human gut microbiome whilst separating this from confounding effects from the host’s physiology, diet, and inflammation, they are not an exact simulant of the human gut and preclinical findings require replication in vivo. Although this study did not include paediatric patients, we have no evidence to suggest that the same effects would not have been observed also in children with IBD on remission. Furthermore, whether the same effects will be observed in patients with active Crohn’s disease, where dysbiosis tends to be more profound, should be explored in future research.

The current study demonstrated that despite extensive microbial dysbiosis, patients with IBD have a similar capacity to break down fibre and release SCFA as healthy people. Fibre supplementation alone is unlikely to restore to a healthy status the compositional dysbiosis characteristic of the microbiome of patients with IBD. Future clinical studies using healthy and IBD human populations are required to confirm the in vitro results presented here and whether these would be helpful in supporting future nutritional interventions and determining the medical importance of the finding to humans.

## 5. Conclusions

The current study demonstrated that despite extensive microbial dysbiosis, patients with IBD have a similar capacity to break down fibre and release SCFA as healthy people. Fibre supplementation alone is unlikely to restore to a healthy status the compositional dysbiosis characteristic of the microbiome of patients with IBD.

## Figures and Tables

**Figure 1 nutrients-14-01053-f001:**
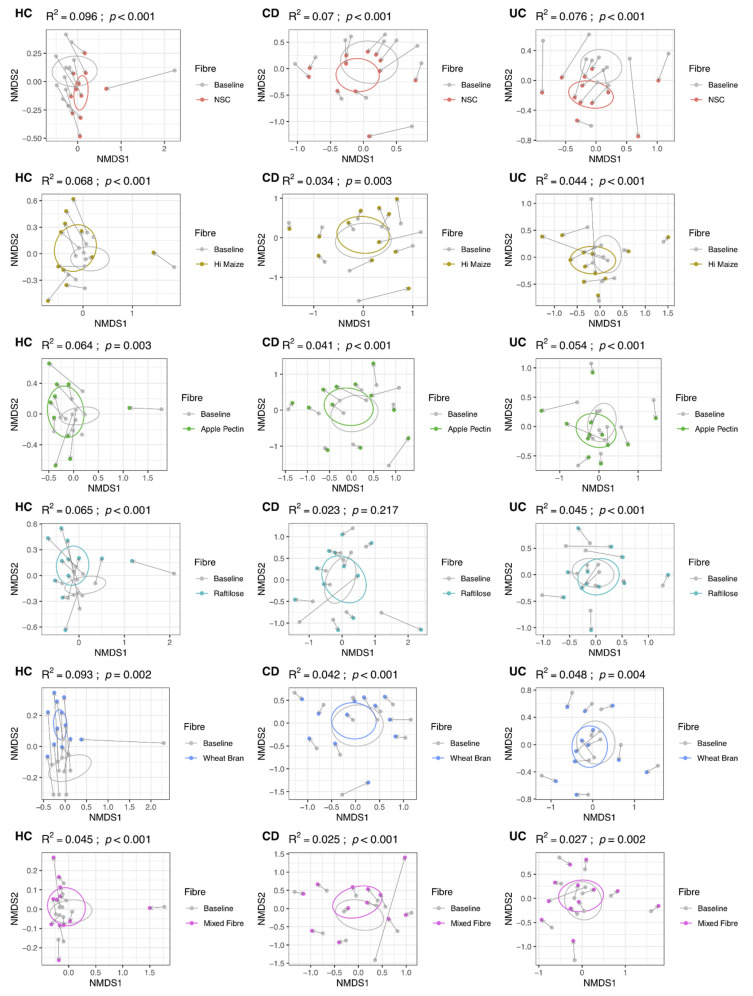
Non-metric multi-dimensional scaling plots, using Bray–Curtis distances, displaying changes in community structure for each fibre substrate and participant group after 48 h fermentation. UC: ulcerative colitis; CD: Crohn’s disease; HC: healthy controls; NSC: non-substrate control.

**Figure 2 nutrients-14-01053-f002:**
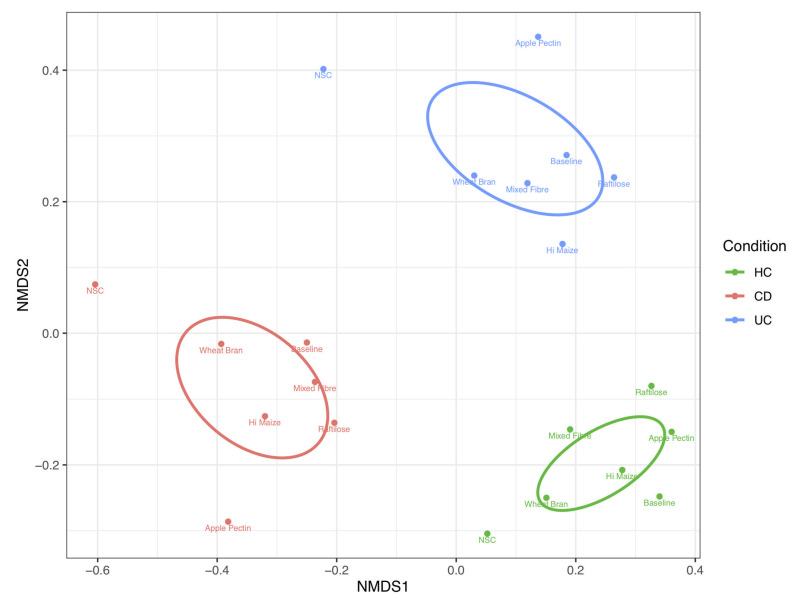
Non-metric multi-dimensional scaling plots, using Bray–Curtis distances for aggregated (mean per fibre substrate) amplicon sequence variant data. Samples were grouped by combined participant group and fibre and the proportional abundances for the amplicon sequence variants were summed. UC: ulcerative colitis; CD: Crohn’s disease; HC: healthy controls; NSC: non-substrate control.

**Figure 3 nutrients-14-01053-f003:**
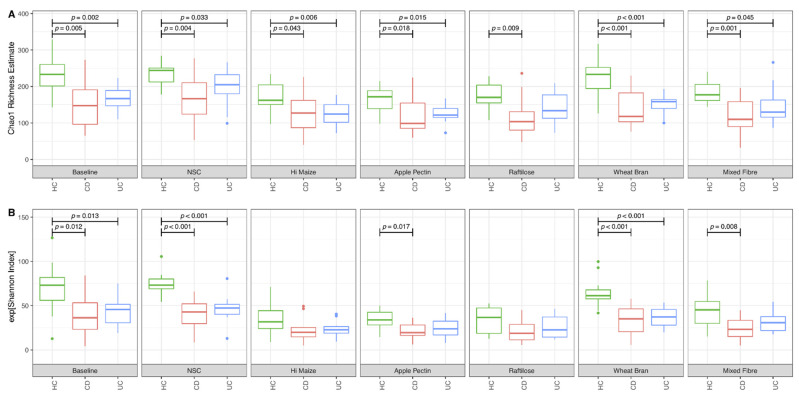
Indices of α diversity and microbiome richness in patients with Crohn’s disease and ulcerative colitis and in comparison to healthy controls prior to and following 48 h fibre fermentation. (**A**) Chao 1 Richness Estimate and (**B**) Shannon Index; UC: ulcerative colitis; CD: Crohn’s disease; HC: healthy controls; NSC: non-substrate control.

**Figure 4 nutrients-14-01053-f004:**
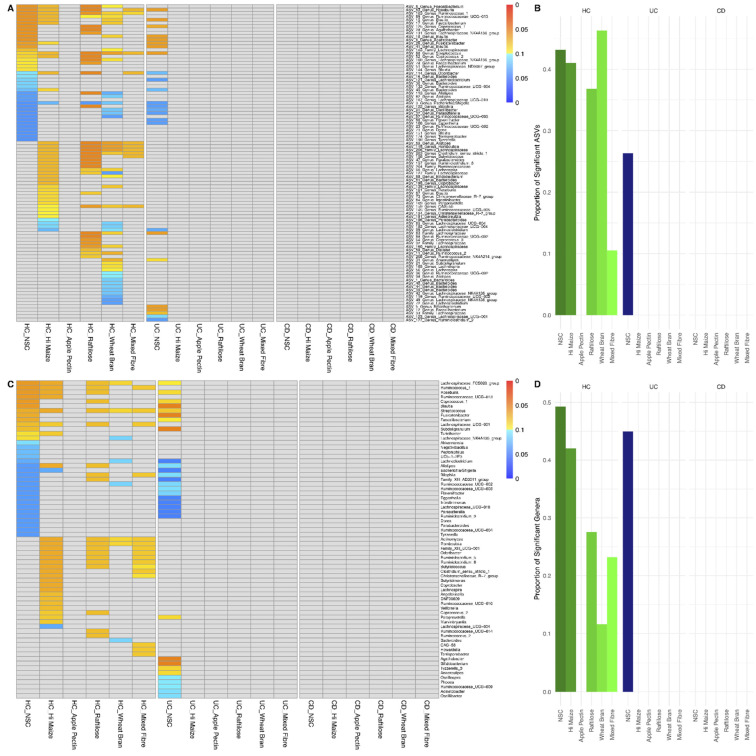
Amplicon sequence variants (**A**) and genera (**C**) that changed significantly after 48 h of fermentation grouped by participant condition and fibre. Blue ASVs significantly decreased in abundance and red ASVs significantly increased. Deeper colour denotes a higher degree of significance. Proportion of total ASVs (**B**) and general (**D**) that changed significantly for each participant group and fibre. UC: ulcerative colitis; CD: Crohn’s disease; HC: healthy controls; NSC: non-substrate control.

**Figure 5 nutrients-14-01053-f005:**
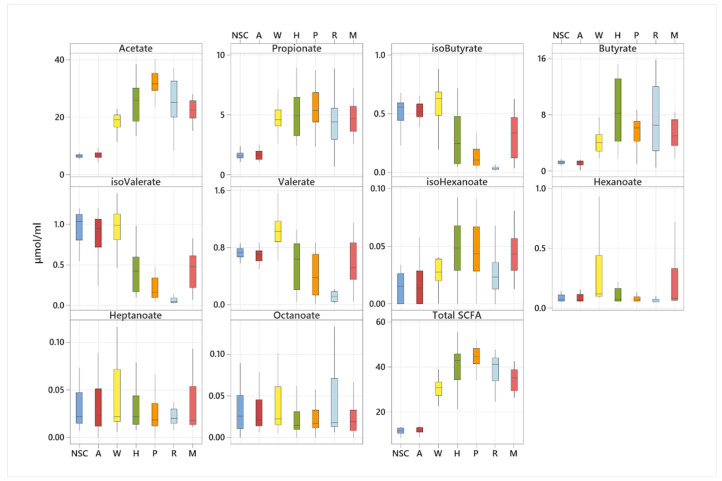
Production of short-chain fatty acids following 24 h fermentation of 5 dietary fibres, a mixed fibre and a no substrate control. NSC: non-substrate control, A: α cellulose, W: wheat bran, H: HI-MAIZE^®^, P P: pectin, R: raftilose, and M: mixed fibre.

**Figure 6 nutrients-14-01053-f006:**
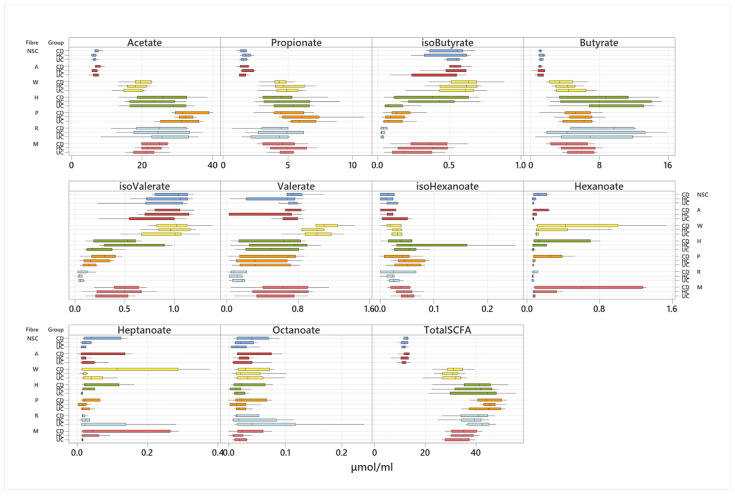
Comparison of short-chain fatty acid production between participant groups, following 24 h fermentation of 5 dietary fibres, a mixed fibre and a no substrate control. NSC: non-substrate control, A: α cellulose, W: wheat bran, H: HI-MAIZE^®^, P: pectin, R: raftilose, and M: mixed fibre.

**Table 1 nutrients-14-01053-t001:** Characteristics of patients with IBD and healthy controls in this study which explored the fibre fermentative capacity of the gut microbiome.

	Microbiome Study			Fermentative Capacity Study		
	Crohn’s Disease (*n* = 12)	Ulcerative Colitis (*n* = 12)	Healthy Controls (*n* = 12)	Crohn’s Disease (*n* = 8)	Ulcerative Colitis (*n* = 8)	Healthy Controls (*n* = 8)
Gender, F/M	5/7	6/6	5/7	5/3	6/2	6/2
BMI, kg/m^2^	25.4 (5.6)	26.0 (5.3)	26.4 (5.8)	25.3 (3.6)	24.7 (3.6)	26.3 (4.9)
Age, years	35.5 (9.0)	50.6 (18.0)	40.0 (13.8)	36.6 (13.9)	45.3 (18.8)	34.3 (10.4)
Disease duration, years	5.7 (7.9)	9.9 (13.3)	n/a	8.7 (4.1)	4.6 (2.1)	n/a
Raised calprotectin, n (%)	5 (42%)	0 (0%)	0 (0%)	1 (12.5%)	3 (37.5%)	0 (0%)
Raised CRP, n (%)	4 (33%)	1 (8.3%)	n/a	0 (0%)	0 (0%)	n/a
Disease location						
Ileocolonic	4 (33%)	n/a	n/a	5 (62.5%)	n/a	n/a
Colonic	3 (25%)	n/a	n/a	1 (12.5%)	n/a	n/a
Ileitis	5 (42%)	n/a	n/a	1 (12.5%)	n/a	n/a
Perianal	0 (0%)	n/a	n/a	3 (37.5%)	n/a	n/a
Disease behaviour						
Inflammatory	8 (67%)	n/a	n/a	3 (37.5%)	n/a	n/a
Stricturing	3 (25%)	n/a	n/a	4 (50.0%)	n/a	n/a
Penetrating	1 (8%)	n/a	n/a	1 (12.5%)	n/a	n/a
Proctitis	n/a	2 (17%)	n/a	n/a	2 (25.0%)	n/a
Left-side	n/a	3 (25%)	n/a	n/a	2 (25.0%)	n/a
Extensive	n/a	7 (58%)	n/a	n/a	4 (50.0%)	n/a

## Data Availability

Data may be available to other authors at request.

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
