# Peer review of "The Effects of Commonly Consumed Dietary Fibres on the Gut Microbiome and Its Fibre Fermentative Capacity in Adults with Inflammatory Bowel Disease in Remission"

_nutrients, 2022, doi:10.3390/nu14051053_

Round 1

Reviewer 1 Report

The author claimed that due to inconsistent results of faecal concentrations of SCFA in IBD patients and exclusive enteral nutrition therapy in children with active Crohn’s disease (CD), they conducted an in vitro evaluation of short chain fatty acids (SCFA) produced in inflammatory bowel disease (IBD) compared to healthy controls (HC).  Five different dietary fibers were fermented as prebiotics, in a small number of adult IBD patients using faecal inoculums.  Nonspecific results of both SCFA concentration (umol/g) by gas chromatography and microbiome by 16S rRNA sequencing, including genera abundance were obtained, and compared with controls.  The authors concluded from the in vitro study that it was unlikely that the dietary fiber supplements could affect the microbiome diversity in IBD patients.

The background summarized general concern regarding the diversity of results in the literature and pointed out the need for clarification by reevaluating the role of pre- and pro-biotics for successful treatment of IBD.  The studies were well designed, and the methods were thoroughly carried out.  The statistics were very sophisticated using ASVs for accuracy in identifying single molecules.  The principal components of PCA seemed easier to visualize on the plot comparing the non-metric multidimensional scaling (NMDS) of ASVs.  The results and discussions were described in depth.

The article is very interesting and important to the clinics caring for IBD patients and to the IBD patients themselves.  However, as the authors stated, that the study was not an exact simulation of human gut physiology for 2 hours of fermentation in vitro was used.  Future clinical studies using healthy and IBD human populations to confirm the in vitro results would be helpful in supporting future nutritional interventions and determining the medical importance of the finding to humans.

Please define: A non-substrate control (NSC) was used for each participant.

Although amplicon sequence variants (ASVs) suggested the accuracy up to single-nucleotide differences, principal component analysis (PCA) is easier for grouping.

Reviewer 2 Report

The manuscript by Konstantinos Gerasimidis et al. invetigated the the effects of dietary fibres on the gut microbiome and its fibre fermentative capacity in adults with IBD. The study is of interest to the readers and I have the following questions and suggestions:

1, The resolution of Figure 4 must be improved. I can hardly read the labelings. The authors must revise. 

2, A Lefse analysis must be conducted to compare the differences of the gut microbiota. 

3, Future research directions must also be discussed. 

4, Why patients with IBD have a similar capacity to break down fibre and release SCFA as healthy people. This mus t be discussed. 

5, The composition of the gut microbiota before fermentation must be provided. This can help to explain why patients with IBD have a similar capacity to break down fibre and release SCFA as healthy people. 

Round 2

Reviewer 2 Report

The authors have revised the mansucript. I suggest to accept it.